# Dietary Supplement Use among Older Cancer Survivors: Socio-Demographic Associations, Supplement Types, Reasons for Use, and Cost

**DOI:** 10.3390/nu14163402

**Published:** 2022-08-18

**Authors:** Harleen Kaur, Teri Hoenemeyer, Kelsey B. Parrish, Wendy Demark-Wahnefried

**Affiliations:** 1Department of Nutrition Sciences, University of Alabama at Birmingham (UAB), Birmingham, AL 35294, USA; 2Department of Health Behavior, University of Alabama at Birmingham (UAB), Birmingham, AL 35294, USA; 3O’Neal Comprehensive Cancer Center at University of Alabama at Birmingham (UAB), Birmingham, AL 35294, USA

**Keywords:** cancer survivors, aging, supplements, vitamins

## Abstract

Most Americans take dietary supplements (DSs) and use is even higher among cancer survivors. This secondary analysis seeks to identify types, reasons, and costs of supplements used by 367 older cancer survivors enrolled in the Harvest for Health vegetable gardening trial and evaluate associations between supplement intake and medical/socio-demographic factors. Descriptive statistics were used to identify supplement type and reasons for use. Average market price was used to estimate cost. Fifty-nine percent of the sample reported supplement use. Female (OR 2.11, 95% CI 1.35–3.30), non-Hispanic White (OR 1.77, 95% CI 1.05–3.0), and breast and gynecological survivors (OR 1.57, 1.03–2.38) were significantly more likely to report DS use compared to males, minorities, and survivors of other cancers. Use of vitamins (39%), multivitamins (23%), and minerals (12%) were the most prevalent. Commonly reported reasons for supplement use were to improve general health (47%) or treat medical conditions (39%) and cancer-related symptoms (12%). DSs daily costs ranged from USD 0.02 to 19.81, with a mean of USD 1.28 ± 1.74, a median of USD 0.78, and a mode of USD 0.34. DS use is prevalent among older cancer survivors, with overall health reported as the leading reason for use. Out-of-pocket recurrent costs can be substantial and underscore the need to promote a nutrient-rich diet whenever possible in this vulnerable population.

## 1. Introduction

Dietary supplement (DS) use is rapidly increasing in the United States (US), with estimated sales of USD 71.8 billion in 2021 and expected growth upwards of USD 128.6 billion by 2028 [1,2]. According to the US Food and Drug Administration (FDA), DSs include vitamins, minerals, herbs or botanicals, amino acids, and other substances [3]. Socio-demographic factors most associated with DS use are: female, older age (≥71 years), non-Hispanic White (NHW) and Asian, non-obese, food secure, higher income, former smokers, and individuals who self-report “excellent” or “very good” health [4]. Additionally, increased use of DSs has been reported previously among cancer survivors [5,6].

The number of cancer survivors is rapidly increasing given improvements in early cancer screening and new therapeutic treatments [7,8,9]. There currently are over 17 million cancer survivors in the United States [7,8,9]; however, these survivors are at increased risk for cancer recurrence, second malignancies, co-morbidities, functional decline, and have specific dietary needs [9,10,11,12,13]. Often, cancer survivors start using DSs after diagnosis and treatment to meet their nutritional needs and improve overall health. It is reported that between 14% and 32% of cancer survivors begin using DSs after receiving initial diagnoses and treatments [14]. Moreover, many adult cancer survivors take DSs without discussing the use with their medical providers [5].

DSs are used to address nutrient deficiencies, and some reports associate their use with improved immune function [15,16]; however, the role of DS to prevent cancer recurrence, second malignancies, and co-morbidities is controversial and has not been established [17]. Moreover, the overuse of DSs has been associated with various nutrient toxicities manifesting in photosensitivity, neurotoxicity, chronic sensory polyneuropathy, bleeding, diarrhea, weakness, blurred vision, and gonadal dysfunction [18,19]. Thus, the World Cancer Research Fund/American Institute for Cancer Research (WCRF/AICR) and US Preventive Services Task Force (USPSTF) guidelines do not recommend the use of DSs for cancer prevention and control [20,21]. Rather, their endorsements focus on following a healthy dietary pattern to achieve adequate nutrition to optimize health outcomes and quality of life [20,21].

However, the use of DSs remains prevalent among cancer survivors, representing USD 6.7 billion in sales annually [5,22]. Types of DSs being taken by cancer patients and survivors vary by geographic location and cancer type, with multivitamins and minerals, vitamins D and B6, magnesium, and fish oil being the most common [5,6,14,22]. Additionally, fear of cancer recurrence is a commonly reported reason for DS use among cancer survivors, even though current guidelines do not support such logic [6,23,24]. A recent investigation reported the estimated cost of supplementation to be USD 1.00 per day among cancer survivors [25]. While fairly modest, this out-of-pocket recurrent cost can be substantial among those living on fixed incomes; this is especially germane for cancer survivors since the average age of a cancer diagnosis is age 66 and over 60% of cancer survivors are age 65 and older [7,8,9].

Understanding the use of DSs is an important step to minimize long-term health risks among older cancer survivors. However, few studies have explored the use of DSs among older cancer survivors (≥65 years), particularly those living in the Deep South, a 5-state region that includes Alabama, Georgia, Louisiana, Mississippi, and South Carolina and represents higher cancer-related health outcomes and higher percentages of minorities who are disproportionately affected by poverty [26]. Utilizing supplement data collected during a recently completed NCI-funded trial, this study aims to characterize supplement use among cancer survivors in the Deep South (identifying types of supplements, the reasons for their use and cost) and then evaluate associations between supplement intake and key medical (cancer type) and socio-demographic factors (sex, race, residency status, education, marital status, and income).

## 2. Materials and Methods

### 2.1. Study Design and Participants

A secondary analysis was performed on baseline data from the Harvest for Health (H4H) study, an NCI-supported randomized controlled trial (RCT) designed to evaluate the efficacy of a home-based vegetable gardening intervention on health outcomes among older cancer survivors across Alabama. Methods of the H4H trial have been published previously and the trial is registered through the NIH ClinicalTrials.gov website (NCT02985411) [27]. H4H participants were screened and recruited between October 2016–February 2021 and were identified from the Alabama Statewide Cancer Registry (ASCR), the University of Alabama at Birmingham (UAB) Registry, and previously established physician referral networks. The UAB Institutional Review Board approved the research protocol (IRB #160328009) and all participants provided written informed consent.

The eligibility criteria for the H4H trial included (1) Medicare eligibility (usually defined as aged 65 years or older); and (2) cancer diagnosis with a 5-year survival rate of ≥60% (i.e., in situ or localized bladder, cervix, gastric cardia, larynx or early-stage multiple myeloma cancer or in situ/loco-regionally staged female breast (male breast cancer cases were not included due to the lower prevalence and lower survival) [28], colorectum, endometrium, kidney/renal, pelvis, oral cavity/pharynx, ovarian, prostate, soft tissue sarcoma, thyroid, melanoma cancers, and all stages of testes cancer, leukemia, and Hodgkin’s or Non-Hodgkin’s lymphoma). Other details regarding the eligibility criteria are published elsewhere [27].

### 2.2. Data Collection and Measures

Data were collected from survey questions on supplement use conducted during the baseline assessment. All participants completed a mailed written survey or an online survey via REDCap^®^ that collected demographic and medical data, such as highest level of educational attainment, marital status, income, and type of cancer treatment received (e.g., surgery, chemotherapy, radiation therapy, immunotherapy, hormonal therapy, and active surveillance). The following information on supplement use was also collected: (1) supplement name; (2) dosage; (3) number of doses; (4) dosage frequency; and (5) reason for use.

### 2.3. Statistical Analysis

This secondary analysis used data on 367 cancer survivors who completed surveys on dietary supplementation. To prepare data for analyses, race/ethnicity was dichotomized as NHW versus other. Primary cancer site was dichotomized as either breast and gynecological cancers or other cancers (e.g., prostate, colorectal, skin, gastrointestinal cancers, and others). Rural and urban residency status was coded using zip-codes by Rural-Urban Commuting Area (RUCA) codes, version 2.0, a Census tract-based classification scheme that uses the Bureau of Census urbanized area, urban cluster definitions, and work commuting data [29,30]. Urban was coded as (RUCA 1.0, 1.1, 2.0, 2.1, 3.0, 4.1, 5.1, 7.1, 8.1, and 10.1) and Rural was coded as (RUCA 4.0, 4.2, 5.0, 5.2, 6.0, 6.1, 7.0, 7.2, 7.3, 7.4, 8.0, 8.2, 8.3, 8.4, 9.0, 9.1, 9.2, 10.0, 10.2, 10.3, 10.4, 10.5, and 10.6), suggested by the USDA and the University of Washington’s Rural Health Research Center coding scheme [29,30]. Other demographic variables such as educational level, marital status, and income were dichotomized as follows: some high school/high school degree vs. some college or above; partnered vs. not partnered; and ≤USD 49,999 vs. ≥USD 50,000 and refused, respectively.

Data on each supplement, the dose, the number of doses and their frequency were used to extrapolate a weekly dose. Given that there is no pre-existing literature on defining the frequency on pro re nata (PRN) use of supplements, a modest assumption of two times per month was made for participants who reported “as needed” for frequency, based on the lowest frequency reported in the sample. Participants who did not report frequencies were excluded from the cost analysis. Supplement costs were explored using Amazon.com (www.amazon.com) accessed between 6 and 13 January 2022. Using methods similar to those previously reported by Xu et al. [31] and Nabavizadeh et al. [32], Amazon product lists were searched for the highest and lowest prices for each supplement, which were then averaged to derive average cost per week. The supplement cost per week of all the supplements reported by individual participants were summed to derive total weekly cost of supplements and then divided by seven to determine total cost of supplements per day.

To explore types of DS reported among older cancer survivors, nine main DS categories were identified based on the supplement name and frequency: (1) vitamins; (2) multivitamin/mineral preparations; (3) minerals; (4) herbals; (5) amino acids/proteins; (6) joint preparations; (7) fatty acids/oils; (8) pre/probiotics; and (9) other. Details on the types of supplements included within each main category are reported in Appendix A.

Reasons for supplement intake were categorized into four groups: (1) for cancer prevention and control, consisting of responses related to preventing cancer recurrence and/or progression; (2) for cancer-related symptom control, with responses related to cancer symptoms; (3) to treat medical conditions, including responses related to medical and chronic conditions such as cardiovascular disease, diabetes, osteoporosis, anemia, vitamin D deficiency, joint and bone conditions, neuropathy, and other medical conditions; and (4) to improve general health, e.g., immunity, inflammation, metabolism, and other general health reasons. To provide more in-depth understanding on common reasons of DS intake for cancer-related symptom control, sub-categories were generated and included (1) hair loss/nails/dry eyes; (2) fatigue/energy; (3) sleep; (4) GI complications; and (5) stress/depression/anxiety. Likewise, medical conditions were sub-categorized into the following: (1) bone health, including vitamin D deficiency; (2) cardiovascular health including hypercholesterolemia/hyperlipidemia; and (3) other health conditions.

The primary thrust of this exploratory analysis was to characterize DS use among older cancer survivors in the Deep South. Thus, descriptive statistics, i.e., averages (as well as other measures of central tendency) and distribution (e.g., standard deviations) were used to relay the data. An exploratory analysis was performed by using the proportion of responses for the types of supplements and reasons for supplement intake. Based on the responses, main categories for types of supplements (vitamins, multivitamin/mineral preparations, minerals, herbals, amino acids/proteins, joint preparations, fatty acids/oils, pre/probiotics, and other) and reasons for supplement intake (prevention control, cancer-related symptom control, medical conditions, and general health) were generated. In addition, after assuring mutually exclusive observations and assumptions of independence, associations between supplement intake (Yes/No) and categorical medical/socio-demographic variables (sex, race/ethnicity, cancer type, residency status, education, marital status, and income) were explored using the chi-square test of association. Statistical analyses for associations were performed using SAS (version 9.4, SAS Institute, Inc., Cary, NC, USA) [33]. To estimate the independent-sample risk, odds ratios and 95% confidence intervals were reported and an alpha level of 0.05 was used to indicate significant associations between supplement intake and medical/socio-demographic variables.

## 3. Results

Sample characteristics of cancer survivors by supplement use are described in Table 1. The average age of cancer survivors was 70 years, and most of the sample was female, NHW, survivors of breast or gynecological cancers, resided in urban areas, had a college degree, a partner, and an earned income of ≥USD 50,000. More than half (59%) of the sample reported taking supplements, with the average daily number of supplements reported as 3.1 ± 2.4.

Table 2 represents the associations between supplement intake and medical/socio-demographic factors. Statistically significant associations were detected between supplement intake and sex, race, and cancer type sub-groups (*p* < 0.05). Female (OR 2.106, 95% CI 1.345–3.296), NHW (OR 1.774, 95% CI 1.051–2.997), and breast and gynecological cancer survivors (OR 1.566, 1.031–2.379) were significantly more likely to take a supplement compared to males, minorities, and survivors of other cancer types. However, no significant associations were detected for residency, education level, marital status, and income.

The five most frequently used supplements were vitamins (39%), multivitamins (23%), minerals (12%), herbals (8%), and fatty acids/oils (8%) (Figure 1). 

Most commonly reported reasons for supplement intake were to improve general health (47%), to treat medical conditions (39%), and cancer-related symptoms (12%), whereas cancer prevention (2%) was the least frequently reported reason for supplement intake (Figure 2).

Bone health (49%) was the most prevalent reason cited under medical conditions, followed by other health conditions (32%) and cardiovascular health (19%) (Figure 3a). Hair loss/nails/dry eyes (38%), fatigue/energy (21%), and sleep (17%) were most commonly reported under cancer-related symptoms with GI complications and stress/depression/anxiety being less prevalent (14% vs. 10%), respectively (Figure 3b).

The daily cost of supplements ranged from USD 0.02 to 19.81, with the mean, median and mode being USD 1.28 ± 1.74/day, USD 0.78/day, and USD 0.34/day, respectively (Table 3). Approximately 57% of older cancer survivors spent USD 1.00 or less on supplements per day and 24% spent between USD 1.01 and 2.00. The remaining 19% spent more than USD 2.00 (Figure 4).

## 4. Discussion

The current study is the first to characterize DS use among older cancer survivors residing in the Deep South, a region that represents a higher proportionate number of cancer cases with poorer survival [34]. This study is also among the few to capture the reasons why cancer survivors take these supplements. Additionally, this supplement report is among only three to perform a cost analysis of supplement use among a heterogenous sample of cancer survivors, and is the first to report supplement costs among those who are elderly (≥65 years or older).

In this current study, more than half (59%) of older cancer survivors reported using DSs. While the proportion of users is much higher than the 40% of users recently reported by Conway et al. in a study of 1049 breast, prostate, and colorectal cancer survivors in the United Kingdom [6], it is far lower than the proportion found among other studies chronicling supplement use among cancer survivors in the US [5,35]. In a nationally representative sample of US adults participating in the NHANES 2003–2016 survey in which there were 2772 cancer survivors and 31,310 cancer-free controls, Du et al. found that 70.4% of cancer survivors reported DS use compared to 51.2% of adults without cancer [5]. A similar proportion, i.e., 74%, was reported by Miller et al. in a substudy on 753 older, long-term cancer survivors screened for the RENEW (Reach-out to Enhance Wellness) RCT [35]. Therefore, it appears that although supplement use among cancer survivors in the U.S. Deep South may be greater than in other countries, within the U.S., it is lower and may be influenced by regional differences. Indeed, a previous national study by Millen and colleagues among the general U.S. adult population using the National Health Interview Survey (NHIS) from 1987, 1992, and 2000 found lower supplement use in the South [36].

Like several other studies [5,6,14,35,36,37], the current study also found that being female and having a breast cancer diagnosis were significant predictors of supplement use. A 2001 report by Conner et al. attributed higher supplement use in females to their role as “custodians of care” and to “greater perception of susceptibility to illness;” however, there is a dearth of data to either confirm or refute such a premise [38]. Our analysis further revealed that NHW survivors were significantly more likely to use supplements compared to survivors of other race and/or ethnic groups. These data corroborate previous findings from NHANES III 1988–94 and NHANES 1999–2000 surveys [39,40], and other reports that have also identified the NHW population as key consumers of supplements [5,36,37,41]; however, it is currently unknown whether race or ethnicity are driving these associations or if higher socioeconomic status among NHWs is the underlying factor [42,43]. Like the reports by Conway et al. and Miller et al. [6,35], the current study did not detect differences in supplement use by education or income; however, these findings differ from larger studies. In analyses of NHANES data, both Du et al. and Chen et al. found education and income as significant predictors of supplement use [5,44], likely due to larger sample sizes and increased diversity.

The older cancer survivors in this study used a wide variety of DSs, with vitamins, multivitamin-multimineral (MVMM), minerals, herbals, and fatty acids/oils being the most prevalent. The three most commonly used supplements reported in this study were vitamins, MVMMs, and minerals; similar to a seminal study conducted by Velicer and colleagues [14]. This study reviewed research published on supplement use between 1999 and 2006 among active cancer patients and long-term survivors and reported that 64% to 81% of their sample used vitamin or mineral supplements, and 26% to 77% used multivitamins [14]. However, reports by McDavid et al. [45] and Du et al. [5] reported greater use of multivitamins or MVMMs followed by vitamins and minerals among cancer survivors, consistent with other studies in this population [35,46]. The use of herbals (8.3%) and fatty acids/oils (7.8%) was also common among cancer survivors in this study. These findings are partially supported by Conway et al. which reported similar use of herbal supplements (9.2%); however, the reported use of fish oil was much higher (13.1%) in this study [6]. Additionally, research by Miller et al. among long-term survivors also found higher use of herbal (18%) and fatty acid/oil (29%) supplements [35].

While previous studies have reported prevalence and types of supplement use, specific reasons for supplement use among cancer survivors are not well understood with few reports available. In this study, “improving general health” was the most commonly reported reason for supplement use. Older cancer survivors are a vulnerable and high-risk population who have multiple comorbidities that affect their overall health [10,11,12]. For example, bone health was the most prevalent reason provided for taking supplements, followed by cardiovascular health. These data are consistent with the report from Du et al., who reported bone health as the third and heart health as the fifth leading reason for supplement use among adult cancer survivors [5]. Osteoporosis is an identified side effect associated with breast cancer and its treatment and is exacerbated by aging, thus justifying the use of calcium, vitamin D and other related bone health supplements [47,48,49]. Additionally, cancer survivors are at a greater risk for cardiovascular disease (CVD) due to cancer treatment-related cardiotoxicity, and the emergence of modifiable CVD risk factors such as hypertension, hyperlipidemia, diabetes, and obesity, not to mention the co-occurrence of underlying CVD in this population [50,51,52,53]. Thus, supplement use in relation to these two common comorbidities is not surprising.

The role of supplement use in cancer control is not well established [17]. Yet, previous studies have identified “prevention of cancer recurrence” as a commonly reported reason for using DSs among cancer survivors [23,24]. However, in this study, cancer prevention was the least frequently reported reason for supplement intake among these older survivors, though symptom management was a key concern. The findings from this study identified that cancer-related symptoms such as hair loss, nail health, dry eyes, fatigue, and sleep were commonly reported reasons for supplement use. Research by Du et al. found that cancer survivors took supplements “to get more energy” and for “eye health” [5], and these data were comparable to the American Cancer Society’s Study of Cancer Survivors-I, which also cited “to give me more energy” and “it’s something I can do to help myself” as common reasons for supplement use [37]. Survivors experience cancer-related sequalae, such as alopecia, fatigue, glaucoma and cataracts, and sleep disturbances during and post-diagnosis, which can impact overall health [54,55,56]. However, most studies among cancer survivors categorize reasons for supplement use between overall motivations to improve general health and prevention of disease [23,24]. Thus, these studies lack specific categories related to controlling cancer-related symptoms, which may pose a greater impact on overall health among older cancer survivors. Therefore, there is a need to develop more refined questionnaires to measure reasons for supplement use among cancer survivors to accommodate recommendations.

Americans spend USD 12.8 billion on supplements annually [57]. However, cost analysis on DSs among older cancer survivors is limited. In this study, the mean daily cost of supplements was USD 1.28 (~USD 460 annually). This is notably higher than an earlier report of complementary and alternative medicine (CAM) use among 2336 cancer survivors that found an annual expenditure of USD 280 on vitamins and minerals [22]. However, the previous study did not include other DS categories such as amino acids, fatty acids, and herbals to define CAM use [22]; hence substantiating the lower supplement cost compared to this study. Data from the current analysis are consistent with a recent study published by Shaver and colleagues using NHANES 2011–2012 data in which the estimated daily cost of supplementation was USD 1.00 per day [25]. Thus, while findings identify fairly modest out-of-pocket recurrent cost among elderly survivors residing in the Deep South, given that many of the respondents were retired and living on fixed incomes, the cumulative costs are nonetheless substantial and could exacerbate financial burden among this vulnerable population.

There are several strengths of this current study. The sample of older cancer survivors in this trial was distributed across Alabama. This study is among the few to categorize reasons and cost among older cancer survivors living in the Deep South compared to previous investigations on supplement use, which tend to be limited to the Northeastern and Western U.S. regions. Unlike other studies that generalize reasons of supplement use to improving overall health and disease prevention, this study was strengthened by providing an in-depth analysis of various reasons for supplement use, particularly cancer symptom control. Furthermore, our study used an open-ended questionnaire to identify supplement use to better understand components of usage, such as dosage, number of tablets, and frequency compared to closed-ended questions.

However, as in all studies, there were limitations. The main shortcoming of this investigation was the smaller sample size, underrepresentation of cancer-types (beyond breast cancer), rural dwellers, and a self-select population of cancer survivors who displayed an interest in participating in a vegetable gardening trial. Moreover, although the income distribution of our sample was low compared to national statistics [58], with 54% of our sample reporting annual household incomes >USD 50,000 compared to 62% of the U.S. population at large [58], it must be borne in mind that residents of Alabama have lower incomes [58]. Hence, the income distribution of the sample was well-aligned with that of the state. However, there was a clear discrepancy between the level of educational attainment within our sample, where the majority (83%) had a bachelor’s or a higher degree as compared to only 33% of the US population [59]. Less dramatic though still a limitation, was the smaller proportion of racial and ethnic minorities which totaled 19.0% of our sample as compared to 24.7% of cancer survivors within Alabama [60]. Additionally, supplement use was self-reported and is subject to measurement error due to respondent and recall bias. To determine the weekly dose for participants reporting frequency values on pro re nata (PRN) use, an assumption of two times per month was made, which may underestimate or overestimate the measurements. Lastly, the cost for each supplement was not reported by participants, and was calculated based on low and high dollar value from amazon.com (Amazon, Seattle, WA, USA). While Amazon provides a global web e-commerce for supplement prices, it is subject to change based on supply and demand, i.e., seasonality changes, region of the supplier, supply chain challenges, quality of the products, sales, and deals and discounts related to Amazon Prime memberships.

## 5. Conclusions

DS use is prevalent among older cancer survivors, with overall health identified as the most common reason for supplement intake. Moreover, out-of-pocket recurrent costs can be substantial among this population. These data reflect that elderly cancer survivors are practicing lifestyle modifications to improve their general health and minimize cancer-related symptoms. This study emphasizes the need to encourage a diet rich in nutrients to minimize daily cost of supplements among older survivors who may live on fixed incomes. Further, survivors should discuss proper use of supplements with their health care providers and individual variability should be considered by physicians when generating supplement recommendations.

## Figures and Tables

**Figure 1 nutrients-14-03402-f001:**
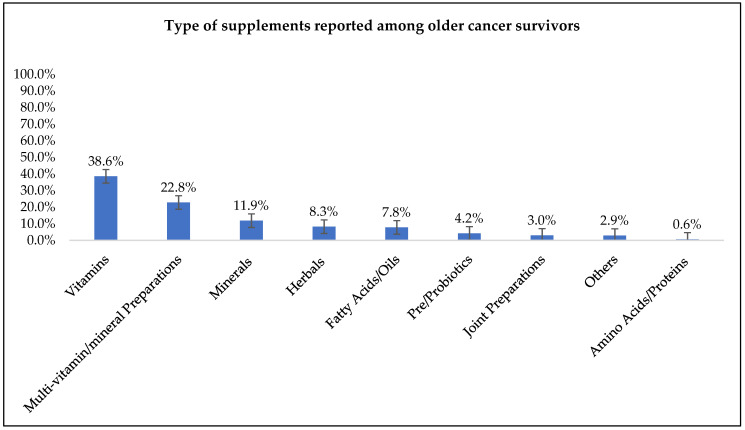
Types of dietary supplements reported among older cancer survivor supplement users (*n* = 215).

**Figure 2 nutrients-14-03402-f002:**
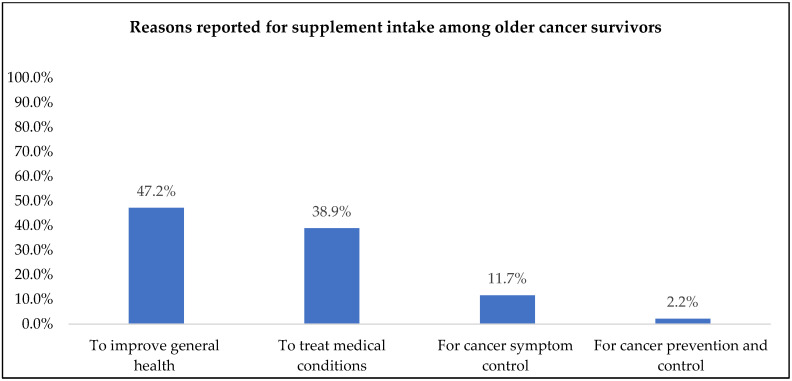
Primary reasons reported for dietary supplement use among older cancer survivor supplement users (*n* = 215).

**Figure 3 nutrients-14-03402-f003:**
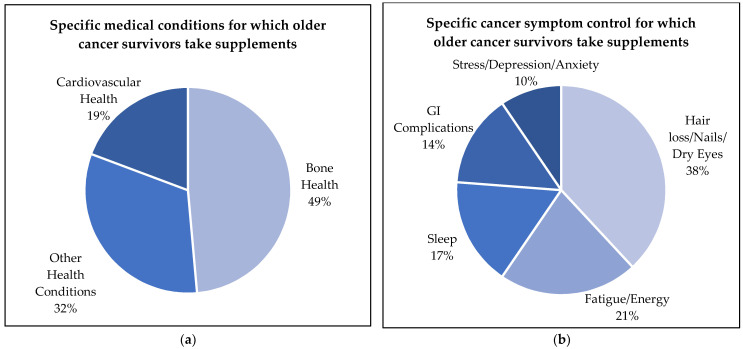
Sub-reasons of dietary supplement intake reported for (**a**) primary medical condition and (**b**) symptom control category among older cancer survivors (*n* = 215).

**Figure 4 nutrients-14-03402-f004:**
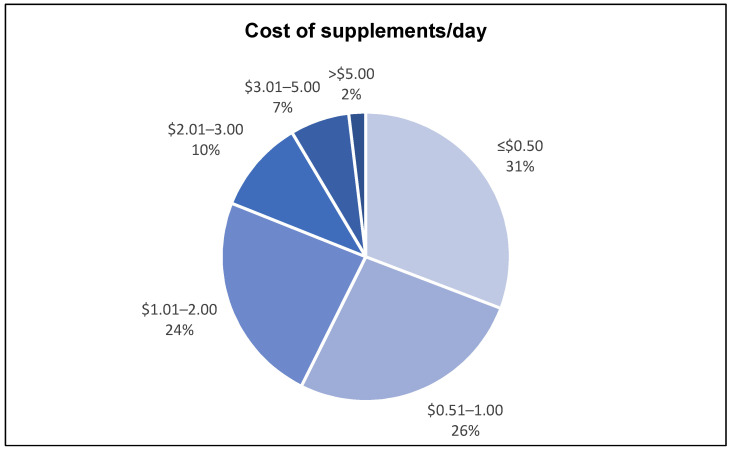
Distribution of supplement cost per day.

**Table 1 nutrients-14-03402-t001:** Sample characteristics of older cancer survivors survey respondents (*n* = 367).

Characteristics	Total (*n* = 367)	Supplement Users (*n* = 215)	Supplement Non-Users (*n* = 152)
Age (years) ^1^	70.0 ± 6.4	70.3 ± 6.5	69.4 ± 6.0
Supplement Intake ^2^	N/A	3.1 ± 2.4	N/A
	*N* (%)	*N* (%)	*N* (%)
Sex			
Female	252 (68.7)	162 (75.4)	90 (59.2)
Male	115 (31.3)	53 (24.6)	62 (40.8)
Race/Ethnicity			
Non-Hispanic White	297 (81.0)	182 (84.7)	115 (75.7)
Other ^3^	70 (19.0)	33 (15.3)	37 (24.3)
Cancer Type			
Breast and gynecological	193 (52.6)	123 (57.2)	70 (46.0)
Other ^4^	174 (47.4)	92 (42.8)	82 (54.0)
Residency			
Urban	324 (88.3)	192 (89.3)	132 (86.8)
Rural	43 (11.7)	23 (10.7)	20 (13.2)
Education Level			
Some HS ^5^/HS degree	60 (16.4)	38 (17.7)	22 (14.5)
Some college or above	307 (83.6)	177 (82.3)	130 (85.5)
Marital Status			
Partnered	228 (62.1)	126 (58.6)	102 (67.1)
Not partnered	139 (37.9)	89 (41.4)	50 (32.9)
Income			
≤USD 49,999	168 (45.8)	104 (48.4)	64 (42.1)
≥USD 50,000 or refused	199 (54.2)	111 (51.6)	88 (57.9)

^1^ Age is represented as the mean and standard deviation. ^2^ Daily number of supplements taken among supplement users is represented as the mean and standard deviation. ^3^ The Other category for race/ethnicity represents non-Hispanic Black (NHB), or survivors who reported more than one race. ^4^ The Other category for cancer type represents prostate, colorectal, skin, gastrointestinal cancers, and other cancers. ^5^ HS = high school.

**Table 2 nutrients-14-03402-t002:** Chi-square test of association between supplement use and medical/socio-demographic variables versus non-use among 367 older cancer survivors.

Variables	Supplement Use (%, *N*)		OR ^1^ (95% CI ^2^)	Chi-Square ^3^	*p*-Value ^3^
Overall (N, %)	**Yes** 215 (58.6)	**No** 152 (41.4)			
Sex (N, %)					
Female	162 (75.4)	90 (59.2)	2.106 (1.345–3.296)	10.7787	0.0010
Male	53 (24.6)	62 (40.8)			
Race/Ethnicity (N, %)					
Non-Hispanic White	182 (84.7)	115 (75.7)	1.774 (1.051–2.997)	4.6658	0.0308
Other ^4^	33 (15.3)	37 (24.3)			
Cancer Type (N, %)					
Breast and gynecological	123 (57.2)	70 (46.0)	1.566 (1.031–2.379)	4.4454	0.0350
Other cancers ^5^	92 (42.8)	82 (54.0)			
Residency (N, %)					
Urban	192 (89.3)	132 (86.8)	1.265 (0.667–2.396)	0.5211	0.4704
Rural	23 (10.7)	20 (13.2)			
Education Level (N, %)					
Some HS/HS grad	38 (17.7)	22 (14.5)	0.788 (0.445–1.396)	0.6670	0.4141
Some college or above	177 (82.3)	130 (85.5)			
Marital Status (N, %)					
Partnered	126 (58.6)	102 (67.1)	0.694 (0.449–1.071)	2.7346	0.0982
Not partnered	89 (41.4)	50 (32.9)			
Income ^6^ (N, %)					
≤USD 49,999	104 (48.4)	64 (42.1)	0.776 (0.511–1.180)	1.4089	0.2352
≥USD 50,000 or refused	111 (51.6)	88 (57.9)			

^1^ OR = odds ratio. The odds ratio estimates are used to identify the magnitude/strength of the association. ^2^ CI = confidence interval. Odds ratio 95% confidence interval that does not include 1 indicates a significant association at the 0.05 alpha level. ^3^ Corresponding chi-square test statistic and p-value indicating associations between supplement intake and medical and demographic variables. ^4^ The Other category for race represents non-Hispanic Black (NHB), or survivors who reported more than one race, or refused. ^5^ The Other category for cancer type represents prostate, colorectal, skin, gastrointestinal cancers, and other cancer. ^6^ Additional income analysis was explored after removing “refused” responses from the income variable. However, no statistically significant associations were detected.

**Table 3 nutrients-14-03402-t003:** Cost analysis of dietary supplements included in primary supplement categories ^1^.

Descriptive	Cost of Supplements/Week	Cost of Supplements/Day
Mean	USD 8.99 ± 12.20	USD 1.28 ± 1.74
Range	USD 0.14–138.67	USD 0.02–19.81
Mode	USD 2.38	USD 0.34
Median	USD 5.46	USD 0.78

^1^ Cost analysis was performed on 211 participants with complete data.

## Data Availability

The datasets generated during and/or analyzed during the current study are not publicly available but may be made available upon reasonable request.

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
