# Peer review of "Dietary Supplement Use among Older Cancer Survivors: Socio-Demographic Associations, Supplement Types, Reasons for Use, and Cost"

_nutrients, 2022, doi:10.3390/nu14163402_

Round 1
Reviewer 1 Report
The manuscript Dietary Supplement Use Among Older Cancer Survivors: Socio-Demographic Associations, Supplement Types, Reasons for Use, and Cost is an interesting socio-demographic analysis of the use of supplements among cancer survivors. However, the limited number of subjects and the choice to use Amazon's prices to define the cost of supplements make the results of this study difficult to compare with other studies.
Although I consider the manuscript worthy of publication some minor considerations should be clarified.
1) In the small sample of cancer survivors (367) analyzed, is the distribution between "Non-Hispanic White" and "Others" representative of the distribution of cancer survivors in the geographical area considered?
2) the authors should detail what they mean by "deep South", also in some passages the authors refer specifically to the State of Alabama, which is the exact geographical area of ​​origin of the subjects participating in the study?
3) how was the $ 49999 limit chosen to define high or low-income individuals? could selecting 3 or 4 income ranges allow to highlight a relationship between supplement consumption and income?
4) from a socio-demographic point of view, does the population sample analyzed differ from the corresponding American population in terms of income, educational level, etc.?
5) are there other studies that have used Amazon prices to estimate the costs of retail products? if so which ones? I also suggest that the authors indicate the time frame in which they monitored Amazon's prices to define the cost of various products.
6) The authors report that the interviewees assert that often the use of supplements is not agreed with their doctor. Is it possible to estimate the effectiveness of using this type of supplement in this particular population? the questionnaire had a question asking if the interviewee thought the use of the supplement was effective?
To conclude, I must say that I appreciated that the authors concluded the manuscript by emphasizing the limitations of their study.
Reviewer 2 Report
Dietary supplements (DS) which are popularly accepted and used among the healthy and cancer survivors, yet there is still unsure information of whether daily intake of DS can benefit the cancer survivors or cause cancer recurrence. Understanding the use of DSs is an important step to minimize long-term health risks among older cancer survivors. Article nutrients-1839819 by Dr. Kaur had investigated the using of dietary supplements (DS) among older cancer survivors. Information of four key factors, were collected via descriptive statistics, which including the main populations, reasons for uses and the types of supplements, as well as the overall cost of daily intake. The study aimed to provide useful information for promoting the accessibility of DS to the vulnerable population.
1. All data in through-out of text of abstract are presented in mean%, all data should be provided as mean±std or other formats to show the statistical analysis.
2. This study are conducted with the human as objects, which the ethical statement is required.
3. In table 1, all data are presented in mean±std, the same should be done in the main text, i.e. in line 174, etc.
4. In Figure 1, please add the error bar for every set of data.
